# E-Beam Irradiation and Ozonation as an Alternative to the Sulphuric Method of Wine Preservation

**DOI:** 10.3390/molecules24183406

**Published:** 2019-09-19

**Authors:** Magdalena Błaszak, Agata Nowak, Sabina Lachowicz, Wojciech Migdał, Ireneusz Ochmian

**Affiliations:** 1Department of Chemistry, Microbiology and Environmental Biotechnology, West Pomeranian University of Technology in Szczecin, Słowackiego 17 Street, 71-434 Szczecin, Poland; blaszak.magdalena@zut.edu.pl (M.B.); agata.nowak@onet.eu (A.N.); 2Department of Fermentation and Cereals Technology, Wroclaw University of Environmental and Life Sciences, Chełmońskiego 37 Street, 51-630 Wrocław, Poland; sabina.lachowicz@upwr.edu.pl; 3Institute of Nuclear Chemistry and Technology, 16 Dorodna Street, 03-195 Warsaw, Poland; w.migdal@ichtj.waw.pl; 4Department of Horticulture, West Pomeranian University of Technology in Szczecin, Słowackiego 17 Street, 71-434 Szczecin, Poland

**Keywords:** polyphenols, color, yeast, wine quality, wine preservation

## Abstract

Potassium metabisulphite is usually used for microbial stabilization in the process of vinification and wine preservation, but it is considered to be allergenic. The objective of the present study was to assess the efficiency of ozonation and ionizing radiation as alternatives to wine sulphurization. The efficiency of yeast removal and the retention of the chemical quality of wine were evaluated. Wine was subjected to 60 min of ozonation, and radiation doses were set at 1–10 kGy. Moreover, a combination of ozonation and ionizing radiation treatment was used. The ozonation of wine did not produce the expected results. That is, it did not limit the number of yeast cells. From the sixth minute, a significant deterioration in the taste and the color of the wine was found. Ionizing radiation at a dose of 1 kGy reduced the yeast count by 95.5%, and a reduction of 99.9% was seen after the application of 2.5 kGy. Moreover, these doses did not have a significant effect on the organoleptic properties or the chemical composition of wine. The total amount of polyphenols reduced from the maximum of 1127.15 to 1023.73 mg at the dose of 5 kGy. Radiation is widely used to preserve food products. Its use for finished wine preservation may be an alternative to sulphurization.

## Highlights

Ozonation does not eliminate yeast, and the quality of wine decreases;

Sulphurization eliminates yeast but reduces the quality of wine;

Electron beam irradiation eliminates yeast and slightly affects the quality of wine;

The optimal dose for wine preservation is 2.5 kGy.

## 1. Introduction

Chemical methods are still widely used for food product preservation. They are characterized by low price, simplicity, and efficiency. However, manufacturers are still searching for new methods of product preservation. In addition, consumers are demanding products without chemical additives [1,2,3,4]. This is certainly influenced by the increasing societal awareness and knowledge of environmental and food pollution that contributes to increased incidence of cancers and allergies. It is increasingly common to focus more on the chemical composition of products, and in particular on artificial chemical additives (i.e., E symbols) [1,5]. Sulphur compounds that are usually used for wine preservation and in the wine production process include E220 (sulphur dioxide), E222 (sodium hydrogen sulphite), E223 (sodium pyrosulphite), E224 (potassium metabisulphite), and E228 (potassium hydrogen sulphite) [6,7]. The authorized food additives with the nature of preservatives and antioxidants are listed in the amended Regulation (EC) No. 1333/2008 of the European Parliament and of the Council of 16 December 2008 on food additives (OJ L354, p16, 31/12/2008) [8,9]. Upon verifying the list of additives, The Scientific Committee on Food or the European Food Safety Authority (EFSA) amends its original positive evaluation for some of them. Repeated evaluation of food additives will be carried out until the end of 2020 [10]. The permissible level of sulphur dioxide in wine is 150–400 mg/L. In wine, increases in the level of residual sugar increase the risk of undesirable secondary fermentation, and thus, the level of sulphur dioxide is higher. Sulphur compounds are present in much greater amounts in spices and dried fruits (1000–2000 mg/kg). Attention should also be paid to the variable resistance of microorganisms to sulphur compounds [6].

Winemaking yeast such as *Saccharomyces cerevisiae* is the most commonly used species for the alcoholic fermentation of grape musts. *Saccharomyces bayanus* and *Schizosaccharomyces pombe* are used to degrade malic acid [11]. Moreover, grape musts contain a vast biodiversity of microorganisms, including wild yeast of genera *Hanseniaspora*, *Candida*, *Metschnikowia*, *Pichia*, *Rhodotorula*, and *Torulaspora*, as well as bacteria [2]. To maintain suitable values of must and wine process parameters, the sulphurization process is repeated several times; however, the sulphur levels of finished products should not exceed the indicated standards. Thus, sulphites are a commonly found allergenic compound in wines, because sulphur dioxide and its derivatives have been listed as strongly allergenic components by the British Food Standards Agency [12]. In addition, excess sulfites in wine have been linked to health problems in highly susceptible persons, especially patients with asthma [13]. Therefore, further research on alternatives to replace sulphur is needed. Methods that can be used on a large scale include ozonation, UV radiation, pulsed electric fields, magnetic fields, ultrasound radiation, high pressure, and ionizing radiation [14,15,16]. A consensus between the efficient elimination of microorganisms and retaining the qualitative values of wine is also needed [17]. Several researchers have used ozone to preserve foods, but their results are still divergent. The aggressive ozonation of fruits was found to reduce the content of fungi (including yeast) by only 50%. However, a significant decrease in the content of phenolic compounds, anthocyanins, and carotenoids was observed Botondi et al. [18]. In contrast, Artés-Hernández et al. [19] recorded the highest efficiency of ozonation for fungi elimination and for retaining the level of polyphenols in grapes. Moreover, ozone was found to be efficient in restricting the population of *Brettanomyces bruxellensis*, which is a serious issue for the winemaking industry [20]. Thus, the establishment of an efficient ozonation method for individual product groups is important, and both time and dose must be controlled. Ozone is one of the few compounds that has been used in post-harvest food production since 1997. It does not influence food taste, is nondurable, and rapidly decomposes to oxygen. One advantage of ozone is the lack of side products during the rapid 12 h decomposition process to pure oxygen [21].

Electron-beam irradiation is also used for food preservation. The research commissioned by the Food and Agriculture Organization of the United Nations (FAO), the World Health Organization, and the International Atomic Energy Agency has demonstrated that radiation of up to 10 kGy is safe for use in food products [22]. During the ionizing radiation process, pathogenic microorganisms (i.e., bacteria, molds, fungi, and parasites) are killed, and ripening, germination, or senescence processes are inhibited [23,24,25]. The efficiency of ionizing radiation depends on the species or even the strain of microorganisms [26]. Molds such as *Fusarium oxysporum*, *Phytophthora citricola*, *Pythium ultimum*, and *Botrytis cinerea* exhibit sensitivity to irradiation within the range of 1.5–6 kGy [27]. Three strains (932, Ent-C9490, and SEA13B88) of *Escherichia coli* O157:H7 suspended in apple juice were sensitive even to a dose of 1 kGy; however, complete elimination of bacteria occurred after the application of the 2 kGy dose [26]. Spore-forming bacteria of the genera *Clostridium* and *Enterobacteriaceae* were found to be resistant to irradiation; a dose of 4–5 kGy reduced their counts by only 90%. Complete elimination of the abovementioned microorganisms required a dose of at least 10 kGy [28].

It is difficult to produce biologically stable and high-quality wine at a generally acceptable price without the addition of sulphur. Thus, for the purpose of the present study, methods that do not generate high costs were selected. The objective of the present study was to assess the effect of ozonation and e-beam irradiation as alternative methods to wine sulphurization and on the efficiency of yeast elimination without affecting the high quality of wine.

## 2. Results and Discussion

### 2.1. Sulphurization—Influence on Chemical and Organoleptic Properties of Wine and Yeast Viability

The use of potassium metabisulphite at the dose of 100 mg/L on day 15 of fermentation completely eliminated yeast from wine. In the control sample, the yeast count was 4.62 ± 0.04 log_10_ CFU/mL, and a dose of 50 mg/L of sulphur eliminated yeast by approximately 45% with reference to the counts of *S. cerevisiae* in wine not subjected to sulphurization. Thus, a dose of 100 mg/L eliminated 99.9% of yeast (Figure 1). Sulphurization is a basic method used for the decontamination and the preservation of wines, as it is a simple, efficient method that does not generate considerable costs. The permissible level of sulphites in wine is 150–400 mg/L, depending on the type of wine, although much lower doses eliminate microorganisms efficiently [29,30]. Considering the abovementioned advantages of sulphurization, it should be accepted that this method is valid in economic and practical terms; however, the issue of consumer health remains unsolved. Customers’ interest in organic products has led to the search for sulphurization alternatives [13,30].

### 2.2. Ozonation—Influence on Chemical and Organoleptic Properties of Wine and Yeast Elimination Efficiency

In the study of Segovia-Bravo et al. [31], the complete elimination of yeast in the biomass of fermenting olives was obtained with ozone doses of 9–30 g ozone/L (3.63 mg/L at a flow rate of 200 L/h). In vegetable juice, only 14 min of exposure to ozone (3 g ozone/L air at a flow rate of 3 L air/min) eliminated yeast completely [32]. In the present study, the effect of ozonation on yeast viability in wine was examined three times in wines prepared three times from the ground up. In addition, wines at different stages of production were used (i.e., 7, 15, or 30 days). These were preliminary studies which aimed to determine the response of yeast and wine on the intense ozonation. Ozonation at doses of 0.75 g ozone/h and 3.5 g ozone/h and a flow rate of 15 L/min was applied, gradually extending the ozonation time to 60 min. It was noted that ozonation did not eliminate yeast from wine. Independent of the fermentation phase (7, 15, or 30 days of experiment), the exposure time to ozone (2–60 min), the ozonator used, and the variable count of yeast in wine, wine decontamination was not obtained (Figure 2). The use of ozone at a dose of 0.75 g ozone/h had almost no effect on the reduction of the number of yeasts in the wines tested or their quality (5 L of wine were ozonized each time). Therefore, for further research (radiation, ozonation, and the combination of these methods), an ozonator was used at a dose of 3.5 g ozone/h. In the first experiment, a wine cuvée after 30 days of fermentation with yeast (4.05 ± 0.04 log_10_CFU/mL) was subjected to ozonation. After 2 min of ozonation, a decrease in yeast count by approximately 40% was observed with reference to the yeast count in a wine sample not subjected to the process; however, a subsequent increase in the time of ozonation did not greatly increase the lethal effect. To verify the outcomes of the first experiment (A), wine samples were prepared for subsequent experiments (B and C). After 15 days of fermentation, the sample was subjected to 60 min ozonation (experiment B). The longer ozonation process did not reduce yeast counts, and after 60 min of the process, their count was higher by 40% (4.76 ± 0.02 log_10_CFU/mL) than in wine not subjected to ozonation. The ozonation effect may depend on the growth phase of the yeast population in the stationary culture. The cells could be more sensitive at a lower count of active yeast cells and in the phase of slow cell death [30 days of fermentation—experiment (A)]. In some cases, the effect of ozone on the viability of yeast is independent of their elimination but dependent on the modification of the length of individual phases of the life cycle [33]. Najafi and Khodaparast [34] reported that yeast requires more than one hour of ozone exposure to undergo considerable reduction in numbers, and in the present experiment, the longest ozonation time was 1 h. The resistance of the tested strain and the consequent too-short ozone exposure time (or insufficient dose) could be the cause for the low efficacy of the ozonation process on the yeast survival rate. The last experiment (C) was conducted for wine with the highest yeast count (6.46 ± 0.08 log_10_CFU/mL) at day seven of the fermentation process. After the application of 5 min ozonation, no statistically significant changes in yeast counts were observed. Moreover, it was found that ozonation exceeding a 5 min duration (with a dose of 3.5 g ozone/L at a flow rate of 15 L/min) has no practical application. A clear deterioration in the organoleptic and the chemical properties of the wine occurred (Figure 3). Wine tasters assessed how long ozonation could be applied without changing the parameters of the wine. Ozonation contributed to negative changes in wine color in wine prepared from the fruit of “Regent” cultivar. Regent is characterized by an intense red-ruby color. Beginning from the sixth minute of ozonation, changes in wine color were observed as compared to the control sample (Table 1). Measurements performed using a spectrophotometer demonstrated that the greatest change occurred for the L* color parameter (9%). However, considerable changes were also observed in the a* parameter. This change in the level was imperceptible to the human eye. The ozonation process was continued, which resulted in wine discoloration. After 60 min, the L*a*b* color parameters showed a considerable change, and the wine color was close to that of a rose. The degradation of anthocyanins undoubtedly had an effect on this situation. In the subsequent experiments, wine was subjected to 5 min of ozonation. This was the limit to which a change in wine color was not observed. Ozone has a high oxidation potential, and direct contact with ozone can lead to color loss. For this reason, it is usually applied on the surface of post-harvest grapes in order to avoid color loss during/after wine fermentation [35]. Furthermore, the organoleptic examination demonstrated that there were no negative changes in the majority of the assessed parameters after 5 min of ozonation (Figure 3). Quijada-Morin [36] reported significant correlations between sensory determinations and chemical composition.

### 2.3. Irradiation—Influence on Chemical and Organoleptic Properties of Wine and Yeast Elimination Efficiency

All applied irradiation doses (1–10 kGy) significantly reduced the yeast counts in wine subjected to 15 day fermentation (Figure 4). The yeast count in the control wine was 4.81 ± 0.09 log_10_CFU/mL. After exposing wine to a beam of electrons at a dose of 1 kGy, the yeast count decreased by approximately 95% (3.45 ± 0.14 log_10_CFU/mL). A subsequent dose eliminated 99.9% of yeast, and after the application of 5, 7.5, and 10 kGy doses, there were no living cells in wine (Figure 4). Similar results were obtained by examining the effect of e-beam irradiation on the counts of yeast in grape fruit intended for vinification. In the study of Morata et al. [37], the application of low irradiation doses of 0.5 and 1 kGy to fresh grapes significantly reduced the yeast count, but only the dose of 10 kGy eliminated yeast cells completely. They examined the impact of three doses: 0.5, 1, and 10 kGy. The results of previous studies indicate that electron-beam irradiation is very efficient in reducing the yeast and the mold content in agricultural products and food products at 1 and 2 kGy doses [37,38,39]. However, complete elimination of yeast and mold was obtained using higher doses of 5–10 kGy [28,40]. The intensity of food irradiation with electron beams depends on a range of factors (radiation dose, type of experimental material, its consistency, and food storage time from the time of irradiation to laboratory analysis); however, independent of this fact, the experimental results clearly confirm the efficacy of this conservation method.

Moreover, a combination of two factors was tested. That is, wine was first subjected to 5 min ozonation, and after a day, it was exposed to irradiation with a beam of electrons at doses of 1, 2.5, 5, 7.5, and 10 kGy (Figure 4). It was again confirmed that ozonation stimulated yeast development, and electron-beam irradiation was the inhibiting factor for the growth of the tested microorganisms. The applied procedures also had an effect on color change (Table 2). With the increase in irradiation dose, the change in the CIE (Comission Internationale de l’Eclairage) L*a*b* color parameters was amplified in both ozonized and non-ozonized wine. Ozonized wine clearly became brighter in color than the control sample. Degradation of the pigment compounds also occurred, as indicated by the changes in a* and b* parameters and the change in the polyphenol profile. Color changes under the influence of ozone exposure were also observed. During the first hour of ozonation, ozone reduced vinasse color by 87% [41]. The changes were less pronounced in non-ozonized wine. Wine exposed to a 10 kGy irradiation dose became brighter than the control by almost 30%, whereas the parameter describing red color (b*) was reduced by almost two-fold. The changes to a* parameter were minimal. The changes were more pronounced in ozonized wine. Wine subjected to the maximum irradiation level was brighter by 60%. Moreover, considerable changes in the a* parameter were observed, which did not occur in non-ozonized wine. The widely used sulphur compounds also resulted in wine color change. The color is reduced by the double-bonded chains of the phenolic group [42]. However, more intense color was observed in wines prepared from irradiated grapes, in which a considerable increase of anthocyanins content was observed. [37,43]. This may have stemmed from the increased extraction of anthocyanins during maceration. Illumination increases the activity of catechol oxidase, an enzyme associated with the biosynthesis of anthocyanin monomers [44].

#### Polyphenolic Compounds

The UPLC-PDA-QTof-MS/MS technique permitted the detection of the polyphenols profile in the “Regent” cultivar (cv.) wine (Table 3). The qualitative resolution noted a composition with 31 polyphenolic compounds, which were classified into four fractions as 16 anthocyanins, seven phenolic acids, three flavanols, and five flavonols. The first class contained in the “Regent” cv. wine were anthocyanins, including the isomers of three delphinidins, six malvidins, three petunidins, one peonidin, and three cyanidins with MS/MS ions at *m*/*z* = 303, 331, 317, 301, and 287, respectively. Amid the identified compounds were monoglucosides and diglucosides, connections with acetic and *p*-coumaric acids, and acylated isomers. Detected compounds were analyzed by other authors in the must and wine of red and rose grapes [45,46]. Flavonols were depicted by five components, which were qualified as two derivatives as two myricetins ([M−H−162]^−^ at *m*/*z* = 479) and three quercetins ([M−H−162]^−^ at *m*/*z* = 479 and [M−H−176]^−^ at *m*/*z* = 477), in which the loss of the glucose and glucoronoid molecules fragment was recorded. These peaks were detected by other authors in the must and wine of red and rose grapes [46,47]. The flavanols profile in “Regent” cv. wine was depicted by two monomers and one polymer compound. (+)-catechin and (-)-epicatechin were the monomers, with [M−H]^−^ at *m*/*z* = 289. The second compound was a polymer identified as procyanidin B2 with [M−H]^−^ at *m*/*z* = 577 and a fragmentation ion at *m*/*z* = 289 [45,46,47]. The last class of phenols was represented by seven phenolic acids. This group was depicted by two major fractions as hydroxybenzoic acid and hydroxycinnamic acids. The first group was portrayed by gallic acid ([M−H]^−^ at *m*/*z* = 169) with MS/MS at *m*/*z* = 125. The second group was portrayed by fertaric ([M−H]^−^ at *m*/*z* = 325), two coutaric ([M−H]^−^ at *m*/*z* = 295), and two caftaric acids ([M−H]^−^ at *m*/*z* = 311). These compounds were previously confirmed and described by Lambert et al. [46] and Wirth et al. [47] in rose wine.

As many as 31 compounds categorized into four groups were identified in the tested wine (Table 3 and Table 4). The anthocyanin content in wine from the Regent cultivar was higher than in wine made of Tempranillo grapes, which grow under higher temperatures in Spain [37,48]. This is likely a cultivar trait, but the decrease in the anthocyanin content in grape skins under high temperatures (35 °C) could be caused by chemical and/or enzymatic degradation as well as by the inhibition of anthocyanin biosynthesis [49]. Derivatives of five anthocyanins were detected in skins (i.e., malvidin, peonidin, petunidin, delphinidin, and cyanidin). Their content stabilizes 30 days before harvest [50]. The Regent cultivar, in contrast to other cultivars (e.g., Cabernet Cortis—polyphenols 608 mg/L) and bright skin cultivars (74–105 mg/L), is characterized by a very high polyphenol (1860 mg/L) and anthocyanin content (1381 mg/L) [45].

Anthocyanins formed the largest group in the total sum of compounds determined in wine (46%). In red varieties of grapes, the largest group of anthocyanin compounds was derivatives of malvidin: 40.6%–84.9% [51]. Peonidin also constituted a large group of anthocyanins at 22%–44% [52]. In the studied wine made of the Regent cultivar, peonidin comprised only 2% of anthocyanins.

Independent of the process of microbial stabilization in the wine, the amounts of anthocyanins decreased considerably. In the extreme case (wine ozonized for 5 min and irradiated with the highest dose), their content decreased by almost 38% in comparison with control wine. The changes in the amounts of anthocyanins were reflected in color change (Table 2 and Table 4). The highest percentage of degradation (by 66%–67%) was related to cyanidin and petunidin 3-O-coumaroyl-glucoside. In contrast, the highest quantitative change occurred for the malvidin 3,5-diglucoside group of compounds (Table 3 and Table 4). The addition of K_2_S_2_O_5_ further influenced changes in the amounts of anthocyanins (total sulphur dose: 956.26 mg/L; 1/2 dose: 1037.36 mg/L) as compared with control wine (1127.12 mg/L). The applied wine sterilization methods also greatly influenced the amounts of compounds, including among flavan-3-ols. The highest changes occurred in ozonized wine subjected to the highest dose of irradiation (from 65.9 to 16.96 mg/L). Irradiation at a dose of 10 kGy, particularly for ozonized wine, resulted in a significant decrease in the content of catechin and epicatechin (by 83% and 61%, respectively) as compared with control wine. E-beam irradiation influences tannin degradation, and it may simultaneously increase phenolic acid content [53].

Flavonols and total hydroxycinnamic acids and derivatives were found to be relatively resistant to the effect of ozone, irradiation, and sulphur. The irradiation dose of 10 kGy or full sulphur dose had less of an impact on the content of these compounds. Again, the combination of ozonation and irradiation had a negative impact on the content of polyphenol compounds. This study demonstrated that a dose of 2.5 kGy enabled satisfactory wine stabilization, which decreased the polyphenol content to a lower degree and led to an insignificant change in wine color. The changes were less pronounced than those for wine decontaminated with sulphur. Artés-Hernández et al. [54] reported that the post-harvest fumigation of table grapes with ozone greatly increased resveratrol content but reduced anthocyanins. In addition, the dehydration process significantly decreased the content of polyphenols in fruits, and the prolonged treatment with ozone resulted in their further loss [18]. A significant decrease in polyphenol content in wine subjected to ozone treatment was probably caused by oxidation. It is known that ozone is characterized by high values of oxidation–reduction potential, and thus it exhibits strong oxidizing properties [55]. Adamo et al. [56] suggested that the destructive processes of oxidation and γ-irradiation are capable of breaking the chemical bonds of polyphenols, thereby releasing soluble low-molecular-weight phenols. This assumption seems to have been validated by the obtained results. This decrease in polyphenol content may be caused by oxidation and condensation reactions of anthocyanins with other polyphenols and precipitate [57,58]. The changes in the content of polyphenol compounds were also observed in wine subject to UV-C radiation. Under the influence of this type of radiation, the content of anthocyanins decreased considerably, and hence no changes in the content of flavan-3-ols were observed, whereas the level of caftaric acid increased [59].

The change in the content of polyphenols and the color in red wine subjected to irradiation at the dose of 10 kGy was also observed by Morata et al. [37]. The hue of a must indicates its degree of oxidation. Antioxidant concentration in plant cells might also be dependent on the time of evaluation (e.g., immediately after the irradiation treatment or after a certain period of time). Total phenols analyzed in irradiated kale juice immediately after the irradiation were significantly lower than those in the control [60]. However, the phenolic compound level of the irradiated sample became higher than that of the control after day one. This was attributed to the immediate oxidation of the phenolic compounds, thus playing an antioxidant role by reducing the free radicals and the reactive oxygen species induced by irradiation. The decrease in antioxidants is attributed to the formation of radiation-induced degradation products or the formation of free radicals [61]. For example, the irradiation of strawberries at the dose of 1–10 kGy led to the decomposition of phenolic acids (*p*-coumaric, gallic, and hydroxybenzoic acids) [62]. The decomposition of these compounds is assigned to the formation of free hydroxyl (OH•) radicals. The low water content influences the restriction of polyphenol decomposition. The application of the dose of 10 kGy to a dehydrated product did not result in the formation of free radicals [63].

## 3. Material and Methods

### 3.1. Characteristics of the Area of Research and Plant Material

The wine grapes were harvested at research station West Pomeranian University of Technology in Szczecin located in the northwestern part of Poland. The majority of the West Pomeranian Province belongs to the 7A zone on Heinz and Schreiber’s “Map of zones of plant resistance to frost”. However, in Szczecin and in the nearby northern region, minimal temperatures range from −12 to −15 °C, which correspond to values typical of zone 7B. The average temperature during the growing season (April–October) between 1951 and 2012 was 13.7 °C, and rainfall was 391 mm [59]. The soil in the vineyard was an agricultural soil with a natural profile developed from silt-loam, pH 6.9, higher water capacity, and optimal mineral content [64].

The vines were grafted on “SO4” rootstock and planted in 2010 with a north–south row orientation at 1 m × 2.3 m. The vines were pruned with a Guyot (one arm) training system and vertically positioned with eight shoots, and each had two clusters. Other standard vineyard management practices including pest treatment were performed during both growing seasons.

### 3.2. Description of the Variety and Production of Wine

This study involved the dark-skinned vine cultivar “Regent”, which is a German cultivar with increasing interest in its cultivation in cool climate areas. The vine is valued especially due to its high fungus and cold resistance. Berries of the Regent cultivar were harvested in October (25.4 Brix) and immediately crushed in order to prepare grape must. Grape must was inoculated with commercial yeast (*Saccharomyces cerevisiae*; ES181, manufactured and supplied by Enartis Viniarske Potreby s.r.o. The dry active wine yeast (30 g/hL) was prepared with 150 mL of water at 35 °C and added to the grape must.

For preliminary studies that were to assess the effectiveness of ozone in eliminating yeast, wines were fermented for 7, 15, and 30 days. The wine was prepared in accordance with the following methodology in a 25 dm3 stainless steel tank. After filtering, the wine was stored at 18–20 °C. The samples for preliminary ozonation were taken successively.

New wine was prepared for further and comprehensive research. The wine was prepared in glass containers with a volume of 5 dm^3^. The grapes were separated from the stalks, crushed, and then macerated for 7 days at 21 °C. In order to eliminate wild yeast, the must was disinfected immediately after crushing. Potassium metabisulphite at a dose of 50 mg/L was used. Then, after 24 h, the must was inoculated with commercial yeast ES181. Standard yeast medium (Browin 401000) was also added to the must at a dose of 40 g per 100 L. After 7 days of maceration, the must (pulp) was pressed in a wine press. The must (7 days wine) was fermented for the next 7 days and was then filtered (CKP V.4 cardboard filter cartridge). The 15 day wine that was used in the experiment had 12.3% alcohol and 3.5 g/L sugar (FOSS WINE SCANFT 120). The experiment with ozonation and e-beam irradiation of the wine was performed two weeks after the initiation of alcoholic fermentation.

### 3.3. Yeast, Assessment of Their Numbers in the Wine

The test yeast *S. cerevisiae* (ES181, ES Viniarske Potreby s.r.o.) is a widely used species for the vinification of grape musts at an industrial scale. This yeast strain is characterized by high tolerance to alcohol content (16.5%) and sugar content (300 g/L) in the culture medium. Yeast counts in wine were recorded in accordance with the ISO 21527–1:2008 [65]. A specialized yeast culture medium was used (YPG Agar, Sigma-Aldrich). Wine samples after serial decimal dilutions were added to the microbial medium (deep inoculation), incubated for 3 days at 25 °C, and the colony-forming units (CFU) were then counted using the eCount Colony Counter (AllChem). After fermentation and filtering the wine, potassium metabisulphite at the dose of 50 or 100 mg/L was added to the wine. This is the standard, commonly used method of wine preservation. This activity microbiologically stabilizes the wine and additionally preserves it. The dosage is 5–10 g metabisulphite per 100 L of wine. In order to speed up research, the wine maturing process was bypassed. Inoculation was carried out 12 h after the wine decontamination process. The experiments were performed in triplicate.

### 3.4. Ozonation Process

Wines were treated with O_3_ produced by an ozone generator (ZYH 135) under a pump flow of 3.5 g/h and providing 15 dm^3^ of air per minute. O_3_ gas from the discharge generator was introduced directly into a glass beaker filled with wine. The experiment was conducted in a fume cupboard. Sterilization was conducted for 60 min. The 60 min ozonation of the wine was done to check the resistance of yeast to ozone and after what time the wine would change color.

### 3.5. Irradiation

The Institute of Nuclear Chemistry and Technology (Warsaw, Poland) has unique devices and elaborate procedures for the process of irradiation, ensuring high efficiency of sterilization and microbiological decontamination. The Accelerator ELEKTRONIKA 10/10 is a high-power radiation device that allows electron beams with 9 MeV energy and average power up to 10 kW to be obtained. These parameters allow the irradiation process to be performed at commercial scale. The main parameters of the ELEKTRONIKA 10/10 accelerator: pulse electron beam mode; electron energy of 8–10 MeV; average beam power of 10 kW; dose rate in the material ρ ≤ 2.5 g/cm^2^ of about 700 Gy/s [66]. The linear electron accelerator Elektronika 10-10 was used for wine irradiation. The electron energy used for the irradiation of yeasts in wine was 9 MeV. Wine samples were packed in 0.5 L glass bottles and treated with doses in the range 1–10 kGy (in triplicate).

### 3.6. Color Measurement

Color parameters were L* (L* = 100 means white; L* = 0 means black), a* (+*a** means redness; −a* means greenness), b* (+b* means yellow; −b* means blue). Color coordinates were determined in the CIE L*a*b* space for the 10° standard observer and the D 65 standard illuminant. CIE L***a***b*** was measured using a spectrophotometer KonicaMinolta CM-700d [67].

### 3.7. Identification of Phenolic Compounds with the UPLC-PDA-MS/MS Method

Polyphenolic compounds were analyzed using a UPLC-PDA-MS/MS Waters ACQUITY system (Waters, Milford, MA, USA) consisting of a binary pump manager, sample manager, column manager, PDA detector, and tandem quadrupole mass spectrometer (TQD) with electrospray ionization (ESI). The separation was carried out using a BEH C18 column (100 mm × 2.1 mm i.d., 1.7 µm, Waters) kept at 50 °C. For the anthocyanins investigation, the following solvent system was applied: mobile phase A (2% formic acid in water, *v*/*v*) and mobile phase B (2% formic acid in 40% acetonitrile in water, *v*/*v*). For other polyphenolic compounds, a lower concentration of formic acid was used (0.1% *v*/*v*). The gradient program was set as follows: 0 min 5% B, from 0 to 8 min linear to 100% B, and from 8 to 9.5 min for washing and back to initial conditions. The injection volume of the samples was 5 µL (partial loop with needle overfill), and the flow rate was 0.35 mL/min. The following parameters were used for TQD: capillary voltage 3.5 kV; con voltage 30 V in positive and negative mode; the source was kept at 250 °C, and desolvation temperature was 350 °C; con gas flow 100 L/h; and desolvation gas flow 800 L/h. Argon was used as the collision gas at a flow rate of 0.3 mL/min. The polyphenolic detection and identification were based on specific PDA spectra, mass-to-charge ratio, and fragment ions obtained after collision-induced dissociation (CID). The quantitative analysis was based on specific MS transitions in multiple reaction monitoring (MRM) mode. The MRM transitions, the cone voltage, and the collision energy of each individual polyphenolic compound were set manually with a dwell time of at least 25 ms. Before injection, wine samples were filtered through a 0.45 µm pore size membrane filter (Merck Millipore) and injected directly into a chromatographic column. Quantification was achieved by the injection of solutions of known concentrations ranging from 0.05 to 5 mg/mL (*R*^2^ ≤ 0.9998) of phenolic compounds as standards. All determinations were performed in triplicate and expressed as mg/L. Waters MassLynx software v.4.1 was used for data acquisition and processing.

### 3.8. Sensory Evaluation

Wines were subjected to sensory evaluation. A group comprising 35 tasters evaluated the quality of the wine. Before starting the sensory evaluation of wines, the tasters were trained and informed about the purpose of the assessment. The people who made the assessment were not professional tasters. Wine samples (30 mL) were evaluated in 100 mL wine glasses. Color, aroma, taste, acidity, and clarity of wines subjected to ozonation (1–10 min) were evaluated and compared to the parameters with those of control wine (i.e., not ozonated). The test person received 11 wine samples—control and ozonated for 10 min (samples taken every minute). It was assessed whether the time of ozonation perceptibly influenced the wine quality. The evaluators indicated the time from which they observed ozone changes in wine parameters. It was a subjective assessment using only the senses (sight, smell, and taste). The arithmetic mean (from the time of ozonation that did not cause changes, 1–10 min) for each trait of wine quality was calculated on the basis of individual assessments, and a chart was developed.

### 3.9. Statistical Analysis

All statistical analyses were performed with Statistica 12.5 (StatSoft Polska, Cracow, Poland). The data were subjected to one-factor variance analysis (ANOVA). Mean comparisons were performed using Tukey’s least significant difference (LSD) test; significance was set at *p* < 0.05.

## 4. Conclusions

Sulphurization is an efficient yeast elimination method at every stage of vinification. However, considering the allergenic properties of sulphur compounds, the use of alternative methods should be considered.

Ionizing radiation used at moderate doses (1–5 kGy) efficiently eliminated yeast in fermenting wine; moreover, it did not have a considerable effect on the examined organoleptic and chemical properties. Considering that irradiation is widely used to preserve food products, its application for the inhibition of vinification or to preserve finished wine could be an alternative to sulphurization.

Ozonation of fermenting wine did not produce the expected results (i.e., total elimination of yeast by at least 95%). Despite the gradual prolongation of the ozonation process to one hour, in the best case, the yeast population was reduced by half, and their count was the same as that in the control or even higher (depending on the ozonation time and the original count of yeast in the wines).

Ozonation had a negative impact on wine quality. At the applied dose of 3.5 g ozone, from the sixth minute, the degradation of polyphenol compounds occurred—particularly of anthocyanins. Moreover, unfavorable changes occurred in color and organoleptic properties of the wine.

An irradiation dose of 2.5 kGy enabled microbial stabilization of wine. This dose only slightly reduced the content of polyphenols and influenced the wine color to a minor degree. It can be recommended as an alternative to sulphurization-based wine decontamination.

## Figures and Tables

**Figure 1 molecules-24-03406-f001:**
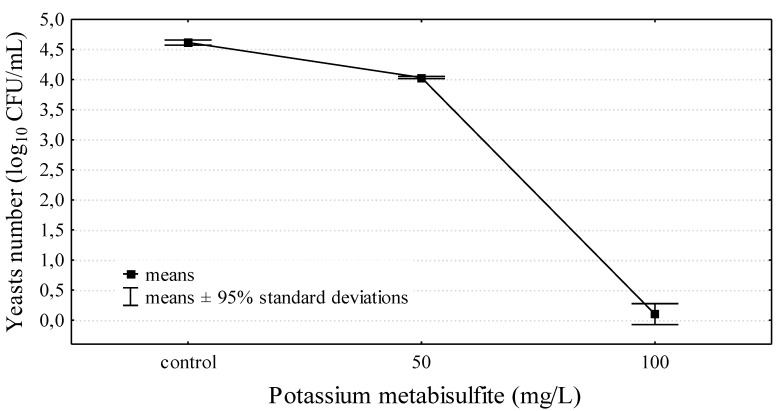
The influence of potassium metabisulfite on yeasts survival in wine fermented for 15 days.

**Figure 2 molecules-24-03406-f002:**
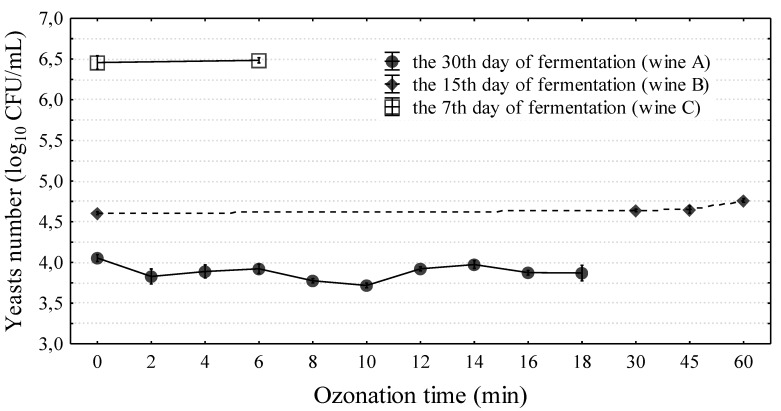
The influence of ozonation process on yeasts survival in wine. There are three separate experiments (wines), marked by letters A–C. Means and 95% standard deviations are marked in Figure.

**Figure 3 molecules-24-03406-f003:**
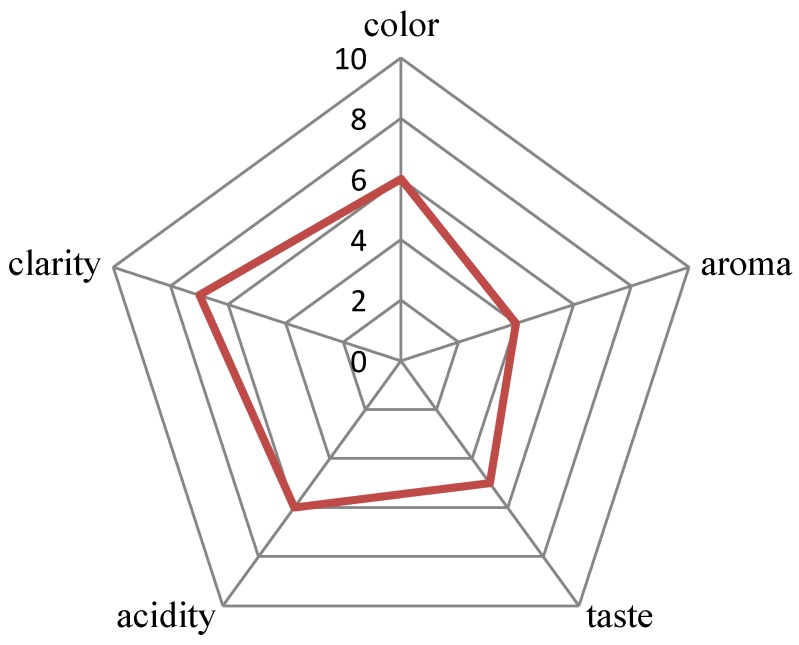
The organoleptic test showing the time of ozonation did not change the organoleptic parameters of the wine (0–10 ozonation time—minute; 0 means control wine—not subject to ozonation. The line indicates the time of ozonation to which no changes in wine parameters were noticed).

**Figure 4 molecules-24-03406-f004:**
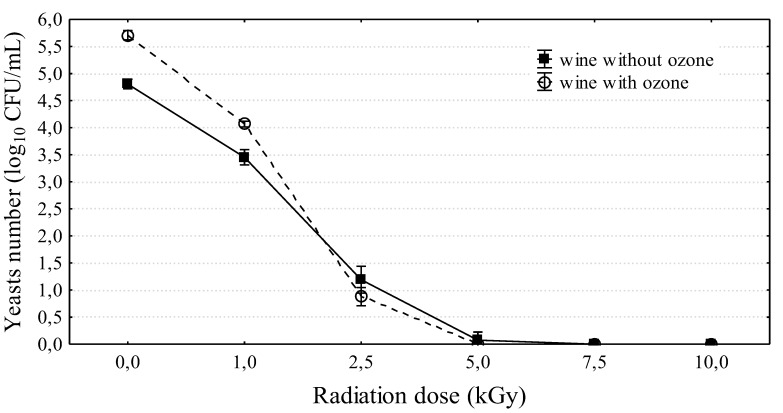
The influence of irradiation doses on yeasts survival in wine (15 day wine). Means and 95% standard deviations are marked in Figure.

**Table 1 molecules-24-03406-t001:** Changes in the color of the wine during the ozonation process.

Ozonation Time	Color CIE		
	L* 100 White; 0 Black	a* Redness	b* Yellow
control—0	32.9 a*	58.4 m	27.5 l
1	33.0 a	58.3 lm	27.5 l
2	33.2 a	57.9 lm	27.6 l
3	34.5 a	57.3 lm	27.4 kl
4	35.1 ab	55.2 kl	27.1 jkl
5	35.7 abc	53.1 k	27.0 jkl
6	37.9 bcd	48.4 j	26.8 jkl
7	38.3 cde	45.0 i	26.3 hijkl
8	39.5 def	43.8 hi	26.0 hijk
9	41.4 fg	40.9 gh	25.7 ghij
10	44.0 g	39.1 fg	25.2 fghi
11	48.2 h	38.4 efg	24.8 efgh
12	50.7 hi	37.5 def	24.4 efg
13	52.6 ij	36.6 def	23.9 ef
14	53.9 j	35.7 de	23.4 e
15	56.6 j	34.3 d	22.8 d
30	62.5 k	29.8 c	17.6 c
45	69.8 l	22.5 b	12.0 b
60	73.6 m	14.7 a	6.4 a

* Means followed by the same letter in columns do not differ significantly at *p* = 0.05 according to Tukey multiple range.

**Table 2 molecules-24-03406-t002:** Changes in the color of wine under the influence of ozonation (5 min), irradiation, and sulphurization.

		Color CIE		
	Irradiation Dose (kGy)	L* 100 White; 0 Black	a* Redness	b* Yellow
without ozone	K	32.6 a*	59.0 ef	26.9 g
	1	34.4 ab	59.0 ef	22.9 f
	2,5	36.3 bc	58.6 ef	19.5 e
	5	37.2 cd	57.2 def	17.2 d
	7,5	39.3 d	56.3 de	15.3 cd
	10	42.2 e	54.5 d	13.0 b
ozone	K	36.5 bc	51.1 c	26.3 g
	1	37.4 cd	42.5 b	25.8 g
	2,5	42.7 e	36.4 a	23.2 f
	5	47.1 f	34.3 a	22.1 f
	7,5	54.1 g	35.9 a	19.1 e
	10	55.0 g	35.6 a	17.9 e
	dose K_2_S_2_O_5_ (mg/L)			
sulfurized	50	38.7 cd	59.5 f	14.8 bc
	100	45.6 f	50.6 c	9.3 a

* for explanation, see Table 1.

**Table 3 molecules-24-03406-t003:** The identification of phenolic compounds of “Regent” wine by retention time (Rt) using their spectral characteristics in ultra-pressure liquid chromatography with photodiode array and mass spectrometry.

Compounds	Rt (min)	MS [M−H]^−^ (*m*/*z*)	MS/MS [M−H]^−^ (*m*/*z*)
Gallic acid	0.87	169	125
Delphinidin 3,5-diglucoside	2.43	627	465/303
GRP—2-S-glutathionylcaftaric acid (cis- and trans- isomers)	2.55	616	
Caftaric acid (cis- and trans- isomers)	2.63	311	179/135
Caftaric acid (cis- and trans- isomers)	2.78	311	179/136
Coutaric acid (cis- and trans- isomers)	2.92	295	163
Cyanidin 3,5*O-*diglucoside	2.93	611	449/287
Delphinidin 3-*O*-glucoside	3.23	465	303
(+)-Catechin	3.38	289	
Malvidin 3,5-*O-*diglucoside	3.63	655	493/331
Coutaric acid (cis- and trans- isomers)	3.80	295	163
Fertaric acid	3.86	325	193/149
Petunidin 3-*O*-glucoside	4.02	479	317
Procyanidin dimer	4.13	577	289
(-)-Epicatechin	4.24	289	
Peonidin 3-*O*-glucoside	4.54	463	301
Malvidin 3-*O*-glucoside	4.72	493	331
Myricetin 3-*O*-galactosode	5.20	479	317
Myricetin 3-*O*-glucoside	5.22	479	317
Delphinidin 3-*O*-(6″-O-acetyl)-glucoside	5.23	507	465/303
Cyanidin 3-*O*-(6″-*O*-acetyl)-glucoside	5.69	491	449/287
Petunidin 3-*O*-(6″-*O*-acetyl)-glucoside	5.86	521	317
Quercetin-3-*O*-glucoside	6.05	463	301
(epi)cat-ethyl-malvidin 3-*O*-glucoside (4 isomers)	6.07	809	357
Quercetin 3-O-glucuronide	6.17	477	301
(epi)cat-ethyl-malvidin 3-*O*-glucoside (4 isomers)	6.33	809	357
(epi)cat-ethyl-malvidin 3-*O*-glucoside (4 isomers)	6.54	809	357
Cyanidin 3-*O*-(6″-*O*-*p*-coumaroyl)-glucoside	7.06	595	287
Petunidin 3-*O*-(6″-*O*-*p*-coumaroyl)-glucoside	7.16	625	317
Malvidin 3-*O*-(6″-*O*-*p*-coumaroyl)-glucoside	7.73	639	331
Quercetin	8.60	301	

**Table 4 molecules-24-03406-t004:** Quantitative determination of polyphenolic compounds in wine depending on the method of microbiological stabilization (mg/L).

Polyphenolic Compounds	Without Ozone						Sulfation		Ozone					
	Radiation Dose						Dose K_2_S_2_0_5_ (mg/L)		Radiation Dose					
	K	1	2,5	5	7,5	10	50	100	0	1	2,5	5	7,5	10
**anthocyanins**														
Delphinidin 3,5-diglucoside	22.65h	21.47g	20.65f	19.50de	17.17bc	16.07a	23.70i	18.03c	19.95de	20.12ef	19.13d	17.31bc	16.05a	16.89ab
Cyanidin 3,5*O-*diglucoside	51.21g	51.61g	48.69f	48.65f	43.04e	39.69d	47.19f	43.03e	42.12e	39.43d	35.27c	32.59b	30.03a	28.29a
Delphinidin 3-*O*-glucoside	27.65e	25.09d	24.50d	24.70d	20.56b	18.90a	24.34cd	18.59a	26.77e	27.84e	24.52d	24.64d	22.98c	21.35b
Malvidin 3,5-*O-*diglucoside	318.24i	313.98hi	307.30h	280.16g	264.53f	242.13d	275.57g	253.53e	258.76ef	252.67e	230.57c	225.44bc	216.06a	219.24ab
Petunidin 3-*O*-glucoside	23.87efg	23.56efg	26.54g	20.85def	15.67ab	12.96a	24.58fg	18.91bcd	20.53cde	19.66bcd	19.89cde	18.03bcd	16.67abc	17.08bcd
Peonidin 3-*O*-glucoside	11.01g	9.75f	9.57f	7.29b	8.84e	7.79bc	8.47de	8.22cd	8.95e	8.93e	8.52de	7.64b	6.28a	6.59a
Malvidin 3-*O*-glucoside	49.23j	48.49j	46.22i	45.47ih	39.64f	34.44c	44.46gh	43.76g	37.65e	36.26de	34.89cd	28.56b	23.30a	24.77a
Delphinidin 3-*O*-(6″-O-acetyl)-glucoside	1.56ef	1.59f	1.47def	1.34cde	1.28cd	1.13bc	1.50def	1.29cd	1.31cd	1.19bc	1.15bc	0.98ab	0.95ab	0.82a
Cyanidin 3-*O*-(6″-*O*-acetyl)-glucoside	1.70ij	1.73j	1.57hi	1.38fg	1.18cd	1.08bc	1.52gh	1.32def	1.46fgh	1.53gh	1.28de	1.19cd	1.02b	0.88a
Petunidin 3-*O*-(6″-*O*-acetyl)-glucoside	2.24g	2.17fg	2.11f	2.01e	1.82bc	1.73ab	1.90cd	1.87cd	1.90cd	1.97de	1.82bc	1.74ab	1.70a	1.77ab
(epi)cat-ethyl-malvidin 3-*O*-glucoside (isomers)	0.90ef	0.88ef	0.91f	0.88ef	0.86cde	0.82bc	0.81bc	0.83cd	0.77ab	0.74a	0.81bc	0.76a	0.73a	0.74a
(epi)cat-ethyl-malvidin 3-*O*-glucoside (isomers)	1.35h	1.31h	1.26g	1.00c	1.02cd	0.86a	1.21f	1.03cd	1.19f	1.22fg	1.05d	1.11e	1.04cd	0.94b
(epi)cat-ethyl-malvidin 3-*O*-glucoside (isomers)	5.45hij	5.49ij	5.39hi	5.53j	5.91h	5.13f	5.34gh	5.22fg	4.86e	4.27cd	4.30d	4.16c	3.93b	3.58a
Cyanidin 3-*O*-(6″-*O*-*p*-coumaroyl)-glucoside	1.12g	1.07fg	1.08fg	1.02ef	0.99def	0.78c	1.06fg	0.94de	0.99def	0.91d	0.72c	0.55b	0.52b	0.37a
Petunidin 3-*O*-(6″-*O*-*p*-coumaroyl)-glucoside	1.55k	1.50jk	1.47j	1.37i	1.31h	1.12f	1.26gh	1.21g	1.15f	1.06e	0.89d	0.81c	0.67b	0.53a
Malvidin 3-*O*-(6″-*O*-*p*-coumaroyl)-glucoside	4.03g	3.96g	3.77f	3.78f	3.43e	3.47e	2.78c	2.49a	3.09d	3.03d	2.85c	2.62ab	2.84c	2.71bc
**total anthocyanins**	**523.77G**	**513.65FG**	**502.49F**	**464.94E**	**427.26D**	**388.10C**	**465.68E**	**420.29D**	**431.45D**	**420.83D**	**387.67C**	**368.12B**	**344.77A**	**346.53A**
**hydroxycinnamic acids and derivatives**														
GRP (cis- and trans- isomers	24.46efg	24.29defg	22.79cd	25.23fg	23.43de	21.53bc	24.14def	25.70g	25.20fg	25.73g	24.88efg	24.20def	20.27b	18.44a
Caftaric acid (cis- and trans- isomers) Tr 2,63	10.84d	11.81ef	12.18f	13.28g	13.43h	13.69h	11.62e	14.32i	9.15c	9.32c	8.94c	9.11c	7.45b	5.71a
Caftaric acid (cis- and trans- isomers) Tr 2,78	0.13c	0.15d	0.15d	0.18e	0.20f	0.17e	0.09b	0.07a	0.15d	0.15d	0.17e	0.13c	0.12c	0.06a
Coutaric acid (cis- and trans- isomers Tr 2,92	238.26de	234.21de	230.48cd	222.95c	213.87b	212.96b	234.13de	212.45b	241.97e	232.21d	234.52de	206.03b	193.80a	197.44a
Coutaric acid (cis- and trans- isomers Tr 3,80	5.03ij	4.96hi	4.86h	4.57g	4.22f	3.40d	5.16j	4.23f	4.50g	4.11f	3.78e	2.87c	1.98b	1.17a
Fertaric acid	93.24f	91.19f	89.02ef	90.44f	76.85b	79.70bc	85.16de	77.06b	84.71d	83.77cd	85.28de	81.54bcd	68.25a	70.13a
**total hydroxycinnamic acids and derivatives**	**371.95F**	**366.61EF**	**359.48EF**	**356.65DE**	**331.99BC**	**331.44BC**	**360.30EF**	**342.83CD**	**365.68EF**	**355.29DE**	**357.57DEF**	**323.88B**	**291.87A**	**292.95A**
**flavonols**														
Myricetin 3-*O*-galactosode	24.55k	24.35k	20.28i	17.66g	17.08f	15.99e	22.11j	19.55h	17.75g	14.28c	14.88d	13.29b	13.04ab	12.65a
Myricetin 3-*O*-glucoside	2.53ef	2.46e	2.26cd	2.60f	2.10b	2.30d	2.19bc	2.48e	2.31d	2.74g	2.58f	2.60f	1.87a	1.93a
Quercetin-3-*O*-glucoside	8.19de	8.11de	8.68f	9.03g	9.58h	12.58j	7.29c	6.37a	9.97i	8.24e	7.58c	7.89d	7.26c	6.88b
Quercetin 3-O-glucuronide	117.40hj	115.22gh	112.57fg	119.20j	124.32k	115.97ghj	115.03gh	104.56c	108.02cde	110.37ef	107.24cde	108.56de	96.72b	84.25a
Quercetin	8.11g	7.94fg	7.47e	9.05h	9.66i	9.21h	7.11cd	7.80f	7.85f	7.92f	7.34de	5.88bc	5.65b	5.24a
**Total flavonols**	**160.77GH**	**158.08FGH**	**151.26DE**	**157.54FGH**	**162.73H**	**156.05EFG**	**153.73EF**	**140.76BC**	**145.89CD**	**143.55BC**	**139.62B**	**138.22B**	**124.54B**	**110.95A**
**flavan-3-ols**														
(+)Catechin	44.95l	43.74l	39.86k	26.15g	23.20f	17.25d	38.19k	34.22j	31.62i	28.44h	19.05e	14.72c	12.18b	7.50a
Dimer B2	9.21h	9.34h	7.18e	6.75d	5.78b	7.23e	8.13g	8.37g	7.74f	7.11e	7.25e	6.48cd	6.20c	4.89a
(-)Epicatechin	11.74j	9.89i	9.02h	6.01d	5.42b	4.54a	7.51f	5.73c	8.25g	8.89h	7.32f	6.90e	4.48a	4.57a
**total flavan-3-ols**	**65.90I**	**62.97I**	**56.06H**	**38.91E**	**34.40D**	**29.02C**	**53.83H**	**48.32G**	**47.60G**	**44.44F**	**33.62D**	**28.10C**	**22.86B**	**16.96A**
Gallic acid	4.76e	4.73e	5.33g	5.69h	4.58e	4.17d	3.82bc	4.06d	4.11d	4.57e	5.04f	3.79b	4.02cd	3.48a
**TOTAL POLYPHENOLS**	**1127.15I**	**1106.05HI**	**1074.61H**	**1023.73FG**	**960.96D**	**908.79C**	**1037.36G**	**956.26D**	**994.74EF**	**968.68DE**	**923.52C**	**862.11B**	**788.06A**	**770.87A**

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
