# Peer review of "E-Beam Irradiation and Ozonation as an Alternative to the Sulphuric Method of Wine Preservation"

_molecules, 2019, doi:10.3390/molecules24183406_

Round 1
Reviewer 1 Report
Dear Editor,
the authors investigated on the antiseptic properties of different concentration of potassium metabisulphite and ozone as well as different irradiation doses in the wine from cultivar Regent. In addition they evaluated the influence of potassium metabisulphite, ozone and irradiation doses on polyphenol contents. They concluded that the better results was obtained by the irradiation at 2.5 kGy, in this case it is guaranteed the microbial wine stabilization with a minimum influence on wine color and polypheno content.
Comments:
In my opinion the chemical and snsorial analyses were perfomed in a short period after vinification while it would have been more interesting to verify what kind of reaction can occur in the bottle or in the barrel during the storage, since the wine is consumed some months after the vinification. The wine is a living element.
The authors should consider that the sulphur dioxide has other properties that improve tyhe wine quality. For these reason it is difficult to replace sulphur dioxide.
Minor comments:
page 2 , line 48, the authors should specify what are E220, 222, 223, 224, and 228;
page 6 line 160 Nitrogen or Ozone? Explain better
Author Response
We want to thank you very much for the valuable comments of the Reviewers. As suggested, the work has been improved in detail. We have responded to all comments and made changes in the text.
We have made great effort to improve our manuscript and we hope that manuscript will be accepted.
Best regards
Magdalena Błaszak and Ireneusz Ochmian
Reviewer 2 Report
The authors of the manuscript "E-beam irradiation and ozonation as an alternative to the sulfuric method of wine preservation" try to offer an alternative to the common method used for wine production. It is true that manufacturers are always searching for new methods of product preservation. Moreover, excess of sulfites in an organism is related to health problems in highly susceptible persons. However, the elimination of microorganisms can decrease the value of wine.
Ionizing radiation used at moderate doses can be an alternative to sulfurization. However, ozonation had a negative impact on wine quality.
I believe the publication of this manuscript can contribute to the knowledge of alternative methods for wine preservation. However, more assays should be performed in the future (out of the scope of this work) to assess the commercial conditions to be applied.
Author Response
Thank you very much for reviewing our manuscript. We made all comments and suggestions in the text. We are conducting further research on the possibility of using radiation in the food industry, especially in winemaking.
Best regards
Magdalena Błaszak and Ireneusz Ochmian
Reviewer 3 Report
The research is interesting and novel. This is good and valuable information. However, there are several spelling and grammatical errors. More information needed in the materials and methods. See attached document for details. Also some of the tables need headings and footnotes to make it more understandable.

Author Response
Thank you for the detailed and substantive evaluation of our manuscript. It influenced the improvement of its quality. We took into account all suggestions and comments contained in the review. We checked the results of alcohol and Brix.
We do not understand the comment: Line 377: "Please recheck alcohol measurement, 12.3% (v / v) from 25.4 ° Brix sugar is highly unlikely." What is highly unlikely: alcohol content or Brix? According to the indicator used by Brix winemakers x the indicator 0.5-0.6 indicates the potential amount of alcohol in wine. We most often use 0.57. Is the Brix level in fruit questionable? In recent years, the average annual temperature has increased significantly in our region. We note several days during the year with temperatures above 30C, at night the temperatures do not fall below 20C. In many varieties, the Brix level exceeds 25.
Best regards
Magdlena Błaszak and Ireneusz Ochmian
Round 2
Reviewer 1 Report
Tha authors did a good work and the manuscript was improved appropriately.
Author Response
Dear Reviewer
Thank you again for the opinion of our manuscript.
The publication has been subject to language correction.
Best regards
Magdalena Błaszak and Ireneusz Ochmian
Reviewer 3 Report
Thank you for making the recommended changes, however, some suggestions and concerns raised in the first report were not addressed. Addressing the issues I pointed out will improve the manuscript and provide clarity to the reader. I also picked up a few new mistakes in the corrected manuscript.
Line 48: Unbold E in E222
Line 70-71: Sentence needs to be rephrased. Consider: In addition, excess sulfites in wine have been linked to health problems in highly susceptible persons, especially patients with asthma [13].
Line 112: Sulphur addition not clearly explained in the materials and methods.
Line 125: The influence of potassium metabisulphite on yeasts survival in 15-day wine or wine fermented for 15 days
Line 134: Remove comma after “i.e.”
Line 167: Not clear in Fig. 3. How does this chart work on which Fig. 3 is based? Are there supposed to be two lines in Fig. 3, one for the control wine and one for wine after ozonation? Or is the control wine scored at 10 for every parameter and a lower score for the ozonated wine indicates a decrease for that attribute? Please provide clarity.
See also comments with regard to line 454.
Line 184: Did Quijada-Morin [36] make the observation that there are significant correlations between sensory determinations and chemical composition? If yes, then consider: Quijada-Morin [36] reported/showed significant correlations between sensory determinations and chemical composition.
Line 187: Table 1 headings: Capital letter needed for “Ozonation”. Use similar CIE headings as in Table 2.
Line 197: Spelling of ozonation
Line 240: Consider excluding Table 4 from this sentence or Table 4 should be numbered Table 3. Tables should be numbered according the order in which they are discussed in the text. Cannot discuss Table 4 before discussing Table 3.
Consider moving Line 239-245 and discussing it under the section where polyphenolic compounds are discussed (i.e Line 294-313).
Line 247: Table 2 needs similar footnotes as Table 1 Also check spelling of sulphurised in the table.
Line 251: Please sort out table numbering. The identification of phenolic compounds of ‘Regent’ wine by retention time (Rt) using their spectral characteristics in ultra-pressure liquid chromatography with photodiode array and mass spectrometry. Remove footnote in the table and at the bottom of the table.
Line 351: The grapes or wine grapes were harvested, not grape fruits!!
Line 379: Remove fruits and replace with grapes or grape berries!
Line 381: Please remove decontamination in this line and throughout the paper. Decontamination has a negative implication. Not all wild yeast or naturally occurring microorganisms are negative. Consider using “preservation”. In line 381, what decontamination process are you referring to? If it is SO2 addition, why not say SO2 was added a concentration of x mg/L or ppm to inhibit wild yeast. If it is another technique than specify. Did you add other nutrients to the must?
Line 381: ….the must was fermented for the next 7 days, and was then filtered (CKP V.4 cardboard filter cartridge). What happened after that. I am trying to understand how you prepared the 30-day wine because you do not specify what happened after day 14. Or does 30-day wine refer to wine that was sampled 30 days after initial inoculation? How and at what temperature were the 30-day wines stored?
Line 383: Please recheck alcohol measurement. How was alcohol measured? A 12.3% (v/v) alcohol measurement from 25.4°Brix sugar is highly unlikely. Sugar conversion ratios usually vary between 0.5 and 0.6. If you use a 0.5 conversion ratio, the alcohol should be around 12.7%. If you use 0.57, the expected alcohol is 14.5%, which is normally the case in wines with such a high initial sugar concentration. What was the alcohol and sugar measurements for the 7-day and 30-day wines, respectively?
Line 385: Consider inserting the sentences from lines 394-396 here: After fermentation (when or what day? Day 14?), potassium metabisulphite at the dose of 50 or 100 mg/L was added to the wine. This is a standard, commonly used wine preservation method.
Line 454-460: More information needed. How were the wines assessed, which method or how were the parameters scored or measured? How does this chart work on which Fig. 3 is based? Please provide clarity.
Author Response
Dear Reviewer
We responded to all comments in the review. We hope that we have resolved all inaccuracies indicated by the Reviewer.
In addition, we would like to clarify that during the production of wine we follow the guidelines of the producer with whom we cooperate. The must must always be disinfected before being inoculated with yeast. We know that wild microorganisms can be useful, but we want the fermentation process to be under control.
The publication was sent for language correction.
Best regards
Magdalena Błaszak and Ireneusz Ochmian
